# Efficient Space–Time Signal Processing Scheme of Frequency Synchronization and Positioning for Sensor Networks

**DOI:** 10.3390/s23042115

**Published:** 2023-02-13

**Authors:** Yung-Yi Wang, Jian-Rung Huang

**Affiliations:** Department of Electrical Engineering, School of Electrical and Computer Engineering, College of Engineering, Chang-Gung University, Tao Yuan 33302, Taiwan

**Keywords:** DOA-CFO estimation, beamforming, MIMO-OFDMA

## Abstract

The orthogonal frequency division multiple access (OFDMA) technique has been widely employed in sensor networks as the data modulation scheme. This study presents a one-dimensional (1D) space–time signal processing scheme for the joint estimation of direction of arrival (DOA) and carrier frequency offsets (CFOs) in OFDMA uplink systems. The proposed approach, initiated by a one-dimensional ESPRIT algorithm, involves estimating the DOAs of the received signal to identify subscriber positions. Spatial beamformers are then used to suppress multiple access interference and separate each subscriber’s signal from the received signal. The outputs of the spatial beamformer are decimated to estimate the CFO of each subscriber. Compared with conventional two-dimensional parameter estimation algorithms, the proposed one-dimensional algorithm has a higher estimation accuracy and significantly lower computational complexity.

## 1. Introduction

Orthogonal frequency-division multiple access (OFDMA) is a multicarrier transmission technique that independently modulates subcarriers within frequencies, achieving prominent performance in combating time-varying fading channel, reducing latency time, and increasing throughput of wireless networks. OFDMA has been employed as the air interface of several wireless connection standards of sensor networks, [1,2].

In OFDMA uplinks, base stations (BSs) have to maintain orthogonality among subchannels [3] through frequency synchronization to avoid multiple access interferences (MAIs). Additionally, BSs require user position information to facilitate the design of transmitting or receiving beamformers for improving transmission throughput [4]. To this end, many two-dimensional (2D) algorithms have been proposed for the joint estimation of carrier frequency offsets (CFOs) and direction of arrivals (DOAs) [5,6,7,8,9,10]. However, most of these algorithms stack the received signal matrices to form high-dimensional snapshot vectors to estimate the two-dimensional parameters, incurring prohibitive computational complexity [9,10].

In order to detect the data symbols from the received signal of the OFDMA uplink system, the BSs have to deal with the multiple access interferences (MAIs) caused by CFOs [11,12,13]. A MAI suppression algorithm for interleaved OFDMA uplinks was proposed in [12] by using the CFO estimate at the BS to construct the receiver weight vector under the minimum mean square error (MMSE) criterion; the resulted filter is referred to as the 1D-MMSE filter. In cooperation with the antenna array equipped to the BS, the 1D-MMSE filter can be extended to a 2D-MMSE filter by using the CFO-DOA estimates of all active users in the network. Although the 2D-MMSE have a better performance in the data detection than the 1D-MMSE, it suffers a high computational complexity. An iterative interference cancellation technique was proposed for distributed MIMO systems [13]. The algorithm can effectively mitigate the MAIs by using the null subspace of the composited channel matrix.

The aforementioned two-dimensional-based estimation algorithms generally exhibit a higher accuracy than the one-dimensional-based algorithms because the former use a high dimensional signal vector contributing a higher processing gain than the later. However, the main disadvantage of conventional two-dimensional-based algorithm is the considerably high computational complexity inherited from the usage of the high dimensional signal vectors. The novelty of this study is to use two consecutive 1D-ESPRIT [11] algorithms in conjunction with the beamforming process to estimate the DOAs and CFOs of the multiuser signals, substantially reducing the computational complexity. To this end, we present a one-dimensional space–time signal processing scheme to jointly estimate the DOAs and CFOs of interleaved OFDMA uplink systems. The proposed approach involves using a 1D-ESPRIT algorithm to estimate the DOA of each user and then accordingly decomposing the received signal into a set of single user signals through a set of spatial beamformers. The output of each spatial beamformer is used to estimate the CFO, which is automatically paired with the leading DOA of the spatial beamformer. According to the CFO estimates, the proposed approach compensates for the CFO and then performs data detection by applying a reduced sized fast Fourier transformation (FFT) on the output of the spatial beamformer, resulting in a significantly lower computational complexity than traditional two-dimensional-based algorithms [8,12].

## 2. System Model

For an interleaved OFDMA uplink system, the transmit signal of user *u* can be expressed by:(1)$$s_{u,n}=\frac{1}{\sqrt{N}}\sum_{k\in \kappa_{u}}{\tilde{S}}_{k}e^{j\frac{2\pi }{N}nk},u=1,\cdots ,U-1$$
where *N* is the total number of subcarriers, ${\tilde{S}}_{k}$ is the data symbol modulated at subcarrier *k*, and $\kappa_{u}={\left\{lL+u\right\}}_{l=0,\cdots ,N^{\prime }-1}$ is the subcarrier index set that the carrier assignment scheme (CAS) assigns to user *u*. $N^{\prime }=\frac{N}{L}$ is the subcarriers available to each user. To avoid inter symbol interference, each user’s signal is inserted with a cyclic prefix (CP) in front of the associated data segment. This study assumes that the BS is equipped with an antenna array of *M*_R_ element. After discarding of the CP, the signal vector received at the BS can be expressed by:(2)$$y_{n}=\sum_{u=0}^{U-1}\left(s_{u,n}⊛h_{u,n}\right)e^{j\frac{2\pi }{N}\varepsilon_{u}n}a\left(\theta_{u}\right)+z_{n},$$
where ⊛ denotes the circular convolution operation, $h_{u,n}$ is the channel response coefficient between user *u* and the BS, $\varepsilon_{u}$ is the CFO of user *u*, $a\left(\theta_{u}\right)$ is the antenna array response vector corresponding to DOA $\theta_{u}$, and $z_{n}$ is the additive white Gaussian noise (AWGN) vector with zero mean and the correlation matrix $E\left\{z_{n}z_{n^{\prime }}^{H}\right\}=N_{0}\delta_{n,n^{\prime }}I_{M_{R}}$. This study assumes a uniform linear array (ULA) comprising of $M_{R}$ half-wavelength-spaced omnidirectional antenna elements, and the response vector is given by:(3)$$a\left(\theta_{u}\right)={\left[\begin{array}{cccc} 1, & e^{j\pi \sin\theta_{u}}, & \cdots , & e^{j\left(M_{R}-1\right)\pi \sin\theta_{u}} \end{array}\right]}^{T}$$
where the superscript ${}^{T}$ denotes the transpose operation. Substituting Equation (1) into Equation (2), the receive signal can be rewritten as:(4)$$y_{n}=\sum_{u=0}^{U-1}\underset{x_{u,n}}{\underset{︸}{\frac{1}{\sqrt{N}}\sum_{k=0}^{N^{\prime }-1}H_{u,kL+u}S_{u,k}e^{j\frac{2\pi }{N}\left(kL+u+\varepsilon_{u}\right)n}}}a\left(\theta_{u}\right)+z_{n}=A\left(\theta \right)x_{n}+z_{n},$$
where $S_{u,k}$ is the data symbol of user, $H_{u,k}=\frac{1}{\sqrt{N}}\sum_{n=0}^{N-1}h_{u,n}e^{-j\frac{2\pi }{N}n}$ is the channel frequency response, $x_{u,n}=\frac{1}{\sqrt{N}}\sum_{k=0}^{N^{\prime }-1}H_{u,kL+u}S_{u,k}e^{j\frac{2\pi }{N}\left(kL+u+\varepsilon_{u}\right)n}$ denotes the CFO-distorted received signal of user *u*, and $A\left(\theta \right)=\left[\begin{array}{ccc} a\left(\theta_{0}\right) & \cdots & a\left(\theta_{U-1}\right) \end{array}\right]$ and $x_{n}={\left[\begin{array}{ccc} x_{0,n}, & \cdots , & x_{U-1,n} \end{array}\right]}^{T}$ denote the spatial response matrix and the received signal vector, respectively. Equation (4) shows that the interleaved OFDMA modulation renders the received signal a quasi-periodic feature in the time domain given by:(5)$$x_{u,n+pN^{\prime }}=\frac{1}{\sqrt{N}}\sum_{k=0}^{N^{\prime }-1}{\overset{⌣}{S}}_{u,k}e^{j\frac{2\pi }{N}\left(kL+u+\varepsilon_{u}\right)\left(n+pN^{\prime }\right)}=x_{u,n}e^{j\frac{2\pi }{L}\omega_{u}p},\;p=0,\cdots ,L-1$$
where $\omega_{u}=u+\varepsilon_{u}$ is the physical frequency bias of user *u*, and ${\overset{⌣}{S}}_{u,k}=H_{u,kL+u}S_{u,k}$ denotes the channel distorted data symbol. Using Equations (4) and (5), we can construct a space–time receive signal matrix by collecting the time samples ${\left\{y_{n+pN^{\prime }}\right\}}_{p=0,\cdots ,L-1}$, and this is given by:(6)$$Y_{n}={\left[\begin{array}{cccc} y_{n} & y_{n+N^{\prime }} & \cdots & y_{n+\left(L-1\right)N^{\prime }} \end{array}\right]}_{M_{R}\times L}=\sum_{u=0}^{U-1}x_{u,n}a\left(\theta_{u}\right)u^{T}\left(\omega_{u}\right)+Z_{n}$$
where $u^{T}\left(\omega_{u}\right)=\left[\begin{array}{cccc} 1 & e^{j\frac{2\pi }{L}\omega_{u}} & \cdots & e^{j\frac{2\pi }{L}\left(L-1\right)\omega_{u}} \end{array}\right]$ is the transpose of the spectral signature vector of user *u*. According to Equation (6), conventional two-dimensional estimation algorithms (e.g., the 2D ESPRIT [8,9], and the 2D MUSIC algorithm [10]) can be invoked to estimate the CFOs and the DOAs. To this end, the received signal matrix is stacked into a vector given by:$${\tilde{y}}_{n}=vec\left(Y_{n}\right)=\sum_{u=0}^{U-1}x_{u,n}\tilde{a}\left(\theta_{u},\omega_{u}\right)+{\tilde{z}}_{n}$$
where $\tilde{a}\left(\theta_{u},\omega_{u}\right)=u\left(\omega_{u}\right)⊗a\left(\theta_{u}\right)$, with ⊗ being the Kronecker product, is the spatial-spectral steering vector of size $M_{R}L\times 1$. Conventional two-dimensional-based algorithms [8,9,10] then use ${\tilde{y}}_{n}$ to calculate the associated autocorrelation matrix and perform eigenvector-decomposition (EVD) to find the subspace matrix to estimate the associated two-dimensional parameters. The large-scale signal vectors provide these two-dimensional-based algorithms a high processing gain and thus ensure high precision in parameter estimation. However, the associated EVD of a high dimensional autocorrelation matrix and the possible searching process give rise to a prohibitive high computational complexity in these two-dimensional-based algorithm.

## 3. The Proposed Method

To mitigate the computational burden, the proposed algorithm executes two one-dimensional ESPRIT algorithms [11] in conjunction with spatial beamforming to estimate the DOAs and CFOs. This study refers the one-dimensional ESPRIT algorithm for the DOA and the CFO estimation as the S-ESPRIT and F-ESPRIT algorithms, respectively.

### 3.1. DOA Estimation

According to (3), the correlation matrix of $y_{n}$ is given by:(7)$$R_{y}=E\left\{y_{n}y_{n}^{H}\right\}=A\left(\theta \right)R_{x}A^{H}\left(\theta \right)$$
where the superscript ^H^ denotes the Hermitian operation, $R_{x}=diag\left\{\sigma_{0}^{2},\cdots ,\sigma_{U-1}^{2}\right\}$, and $\sigma_{u}^{2}=E\left\{{\left|x_{u,n}\right|}^{2}\right\}$ is the correlation matrix of $x_{n}$. For simplicity, we have ignored the noise term in Equation (7). Through EVD, $R_{y}$ can be orthogonally diagonalized and is given by:(8)$$R_{y}=E_{s}\Lambda_{s}E_{s}^{H}+E_{n}\Lambda_{n}E_{n}^{H}$$
where $E_{s}={\left[v_{1},\cdots ,v_{U}\right]}_{M_{R}\times U}$ denotes the signal subspace matrix formed by the U largest eigenvectors of $R_{y}$, $E_{n}$ denotes the noise subspace matrix formed by the rest $M_{R}-U$ eigenvectors, and $\Lambda_{s}$ and $\Lambda_{n}$ are diagonal matrices formed by the eigenvalues corresponding to the eigenvectors in $E_{s}$ and $E_{n}$, respectively. According to Equations (7) and (8), it reveals that $E_{s}$ shares the same column space with A(θ), and it can be expressed by:(9)$$E_{s}=A\left(\theta \right)T$$
where T denotes the associated transformation matrix. The S-ESPRIT algorithm uses the shift invariance property of the signal received by the ULA to estimate the DOAs. To obtain the subarray signals, the S-ESPRIT defines two selection matrices given by:(10)$$J_{1,m}={\left[\begin{array}{cc} I_{m-1}, & 0_{\left(m-1\right)\times 1} \end{array}\right]}_{\left(m-1\right)\times m},\;\mathrm{and}\;J_{2,m}={\left[\begin{array}{cc} 0_{\left(m-1\right)\times 1}, & I_{m-1} \end{array}\right]}_{\left(m-1\right)\times m}$$
where $I_{m-1}$ denotes the identity matrix of size m−1, and $0_{\left(m-1\right)\times 1}$ is a zero vector. Using the selection matrices, two spatial reponse submatrices are defined by:(11)$$A_{1}\left(\theta \right)=J_{M_{R},1}A\left(\theta \right)$$
(12)$$A_{2}\left(\theta \right)=J_{M_{R},2}A\left(\theta \right)$$

Accord to (3), the shift invariance property of $a\left(\theta_{u}\right)$ ensures:(13)$$A_{2}\left(\theta \right)=A_{1}\left(\theta \right)\Psi$$
where $\Psi =diag\left\{\begin{array}{ccc} \psi_{0}, & \cdots , & \psi_{U-1} \end{array}\right\}$ with $\psi_{u}=e^{j\pi \sin\theta_{u}}$. To estimate the DOAs, the S-ESPRIT algorithm constructs two submatrices from the $E_{s}$ in Equation (8), given by:(14)$$E_{s,k}=J_{M_{R},k}E_{s},k=1,2.$$

According to Equation (9), we have:(15)$$E_{s,k}=A_{k}\left(\theta \right)T,k=1,2.$$

From Equation (13) and Equation (15), it shows that $E_{s,2}=A_{2}\left(\theta \right)T=A_{1}\left(\theta \right)\Psi T$, and we have:(16)$$E_{s,2}E_{s,1}^{†}=T^{-1}\Psi T$$
where ${}^{†}$ denotes the pseudo-inverse operation. From Equation (16), it is illustrated that ${\left\{\psi_{u}\right\}}_{u=0,\cdots ,U-1}$ are the eigenvalues of $E_{s,2}E_{s,1}^{†}$, and the DOAs can be estimated by:(17)$${\hat{\theta }}_{u}=\sin^{-1}\left(\frac{\arg\left(\psi_{u}\right)}{\pi }\right),\;u=0,\cdots ,U-1.$$

### 3.2. Interference Suppression and Signal Separation

According to the DOA estimates, the proposed algorithm separates the signal of each user from the received signal in Equation (2) through a spatial beamformer steered at the leading DOA ${\hat{\theta }}_{u}$. The associated beamforming weight vector is designed under the minimum variance distortionless response (MVDR) criterion given by:(18)$$w_{u}=\arg\underset{w}{\min}E\left\{{\left|w^{H}y_{n}\right|}^{2}\right\},\;s.t.\;A^{H}\left(\hat{\theta }\right)w=e_{u+1},\forall u=0,\cdots ,U-1$$
where $e_{u}$ denotes the elementary vector of size U×1 that takes 1 as the *u*th element and 0 elsewhere. The constraint $A^{H}\left(\hat{\theta }\right)w=e_{u+1}$ illustrates that the spatial beamformer of user *u*, which can retain the signal led by $a\left({\hat{\theta }}_{u}\right)$ and suppress the MAIs from the DOAs ${\hat{\theta }}_{u^{\prime }},\forall u^{\prime }\neq u$. Direct manipulations yield:(19)$$w_{u}=R_{y}^{-1}A\left(\hat{\theta }\right){\left[A^{H}\left(\hat{\theta }\right)R_{y}^{-1}A\left(\hat{\theta }\right)\right]}^{-1}e_{u+1}.$$

The output signal of the spatial beamformer is given by:(20)$$y_{u,n}=w_{u}^{H}y_{n}=x_{u,n}+{\tilde{z}}_{u,n},\;\forall u=0,\cdots ,U-1$$
where ${\tilde{z}}_{u,n}$ denotes the associated output noise. According to Equation (20), the proposed algorithm can estimate the CFO of user u and automatically pairs the resultant CFO estimate with the leading DOAs of the associated beqmformer.

Practically, the correlation matrix $R_{y}$ can be implemented by the sample-averaged counterpart given by:(21)$${\hat{R}}_{y}=\frac{1}{N}\sum_{n=0}^{N-1}y_{n}y_{n}^{H}.$$

In addition, the inverse of ${\hat{R}}_{y}$ can be implemented by using the eigenspace counterpart:(22)$$R_{y}^{-1}=E_{s}\Lambda_{s}^{-1}E_{s}^{H}+E_{n}\Lambda_{n}^{-1}E_{n}^{H}$$

By using Equation (21), the weight vector in Equation (19) can be expressed by:(23)$$w_{u}=E_{s}\Lambda_{s}^{-1}E_{s}^{H}A\left(\hat{\theta }\right){\left[A^{H}\left(\hat{\theta }\right)E_{s}\Lambda_{s}^{-1}E_{s}^{H}A\left(\hat{\theta }\right)\right]}^{-1}e_{u+1}.$$

### 3.3. CFO Estimation

In order to estimate the CFOs, the proposed algorithm decimates the output of each spatial beamformer in Equation (20) and forms the signal vector of user u given by:(24)$${\bar{y}}_{u,n}={\left[\begin{array}{c} y_{u,n}\\ y_{u,n+N^{\prime }}\\ \vdots \\ y_{u,n+\left(L-1\right)N^{\prime }} \end{array}\right]}_{L\times 1}=x_{u,n}u\left(\omega_{u}\right)+{\bar{z}}_{u,n},n=0,\cdots ,N^{\prime }-1$$
where we have used the quasi-periodic feature in Equation (5), and ${\bar{z}}_{u,n}$ is the associated noise vector. Accordingly, the frequency bias can be estimated through the F-ESPRIT algorithm, which is a simplified version of the ESPRIT applying for a single-user scenario. The associated correlation matreix for user u is given by:(25)$${\bar{R}}_{u}=\frac{1}{N^{\prime }}\sum_{n=0}^{N^{\prime }-1}{\bar{y}}_{u,n}{\bar{y}}_{u,n}^{H},\;u=0,\cdots ,U-1$$

Table 1 summarizes the F-ESPRIT algorithm.

According to the physical frequency estimate ${\hat{\omega }}_{u}$, the CFO of user *u* can be estimated by ${\hat{\varepsilon }}_{u}={\hat{\omega }}_{u}-\hat{u},$, where $\hat{u}$ is the subcarrier index estimate of user *u*, which is obtained from the integer nearest to ${\hat{\omega }}_{u}$. Because ${\hat{\varepsilon }}_{u}$ is estimated from the output signals of the spatial beamformer *u*, it can be automatically paired with the associated leading DOA ${\hat{\theta }}_{u}$, without extra pairing processes.

To decode data, the proposed algorithm uses ${\hat{\omega }}_{u}$ to form the spectral signature vector $u\left({\hat{\omega }}_{u}\right)$, and then it compensates for the frequency bias in ${\bar{y}}_{u,n}$ for frequency synchronization. From Equations (4) and (24), the resulting signal is given by:(26)$${\hat{x}}_{u,n}=u^{H}\left({\hat{\omega }}_{u}\right)e^{-j\frac{2\pi }{N}{\hat{\varepsilon }}_{u,n}}{\bar{y}}_{u,n}=\frac{1}{\sqrt{N}}\sum_{k=0}^{N^{\prime }-1}{\overset{⌣}{S}}_{u,k}e^{j\frac{2\pi }{N}\left(kL+u\right)n}+{\tilde{z}}_{u,n}$$

Consequently, the data of user *u* can be decoded from ${\hat{x}}_{u,n}$ through a $N^{\prime }$ -point FFT given by:(27)$${\hat{S}}_{u,k}=dec\left\{\frac{\frac{1}{\sqrt{N^{\prime }}}\sum_{k=0}^{N^{\prime }-1}{\hat{x}}_{u,n}e^{-j\frac{2\pi }{N}\left(kL+u\right)n}}{{\hat{H}}_{u,kL+u}}\right\}$$
where ${\hat{H}}_{u,kL+u}$ is the associated channel response estimate, which is assumed known to the BS.

### 3.4. Computational Complexity

The proposed algorithm requires $\left(NM_{R}+UN^{\prime }L\right)$ flops for the calculation of the sample averaged correlation matrices in (21) and (25), $12\left(M_{R}^{3}+UL^{3}\right)$ flops for the determination of EVD of the correlation matrices [14], $2M_{R}^{2}+M_{R}^{2}U^{2}$ flops for the calculation of the beamforming vector in (23), and $UN^{\prime }\log_{2}N^{\prime }$ flops for the calculation of the FFT in Equation (27). Therefore, the complexity of the proposed algorithm is $12\left(M_{R}^{3}+UL^{3}\right)+M_{R}^{2}\left(U^{2}+2\right)+\left(NM_{R}+UN^{\prime }L\right)+UN^{\prime }\log_{2}N^{\prime }$. On using a grid-size of $10^{-q}$, the computational complexity of the methods reported in [10] was approximately $12M_{R}^{3}L^{3}+\left(N+10^{q}\pi \right)M_{R}L+UN\log_{2}N$ flops, and that of the method reported in [8] was $12M_{R}^{3}L^{3}+\frac{4}{3}{\left(M_{R}-1\right)}^{2}{\left(L-1\right)}^{2}+NM_{R}L+UN\log_{2}N$ flops.

## 4. Results and Discussions

Consider an OFDMA system with the following settings: *N* = 512, *M*_R_ = 10, and *L* = 6. Independent Rayleigh fading channels were assumed to exist between each user terminal and the BS. The DOAs and CFOs of the active terminals were randomly selected from the interval $\left|\theta_{u}\right|<\frac{\pi }{2},$ and $\left|\varepsilon_{u}\right|<0.5,\forall u.$ The noise power $N_{0}$ was adjusted to achieve the required signal-to-noise power ratio (SNR).

Figure 1 illustrates the root mean square errors (RMSEs) of the DOA and CFO estimates of the proposed algorithm, the 2D-ESPRIT algorithm [9], and the 2D-MUSIC algorithm [10]. Figure 1a shows that, in addition to having a significantly lower computational complexity, the proposed algorithm possesses a RMSE that is comparable with that of the 2D-MUSIC algorithm for DOA estimation as the SNR increases. The proposed exhibits a marginally lower error than the 2D-ESPRIT algorithm for CFO estimation, as illustrated in Figure 1b.

According to the DOA and CFO estimates obtained in Figure 1, Figure 2 illustrates the comparisons of the symbol error rates (SERs) of the proposed 1D MAI suppression, minimum mean square error (MMSE) [12], and 2D-MMSE (ST-MMSE) algorithms [12] under the assumption of known channel response. The number of subcarriers (*N*) was assumed to be 128 and 512 for assessing the performance of MAI suppression with respect to different OFDMA symbol sizes. For both values of *N*, the proposed algorithm has lower SERs than the other two methods because the proposed constrained optimization strategy in (18) effectively eliminates MAIs according to the DOA estimates. The two conventional MMSE-based algorithms require a sufficiently large number of stacked snapshot vectors to reduce the estimation error of the associate covariance matrix. Therefore, the SERs of the conventional MMSE-based algorithms decrease when the number of subcarriers decreases, as shown in Figure 2.

Figure 3 presents a comparison of the computational complexity of the proposed algorithm with that of the two-dimensional-based algorithms in [8,10]. The scale factor of the 2D-MUSIC algorithm is q=4. The number of antennas is in the range from $M_{R}=4$ to $M_{R}=20$, and the number of subcarriers N=512. Both conventional two-dimensional-based algorithms, [9,10], use a high dimensional array vector for the parameter estimation, resulting in high computational complexity as discussed. Figure 3 demonstrates that the proposed one-dimensional-based algorithm has a substantially lower complexity than that of the other two algorithms. We thus conclude that, in addition to having improved performance in data detection (as indicated in Figure 2), the proposed algorithm is more computationally efficient than are the conventional two-dimensional-based algorithms.

## 5. Conclusions

This study presents a low-complexity user positioning and frequency synchronization method for interleaved OFDMA uplinks. Compared with conventional two-dimensional algorithms, the proposed one-dimensional algorithm effectively mitigates the computational complexity and exhibits superior performance for MAI suppression, leading to at least 3 dB power gain in data detection when SNR > 12 dB. The proposed structured one-dimensional-based algorithm is applicable to other high-dimensional signal processing scenarios, such as the 2D-DOA estimation of the signals received by an uniform rectangular array.

## Figures and Tables

**Figure 1 sensors-23-02115-f001:**
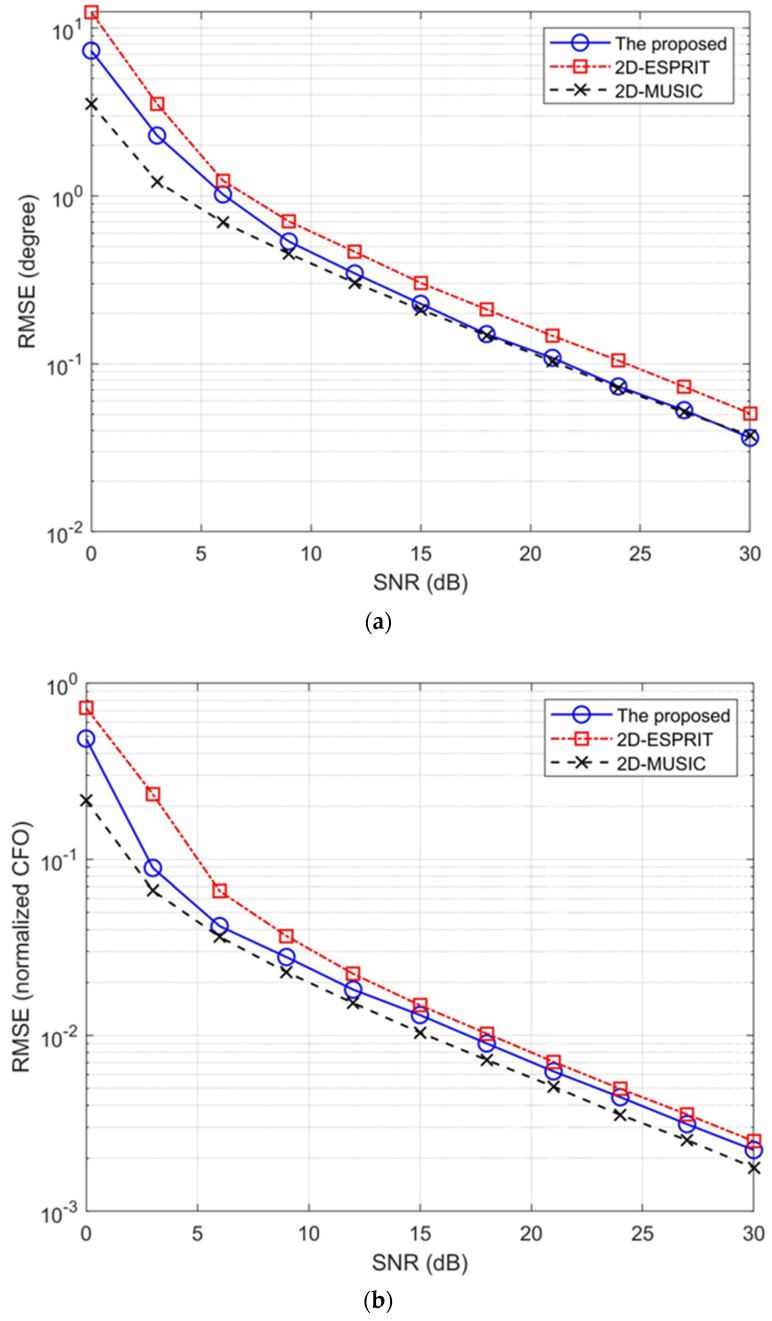
Comparisons of the RMSEs of the proposed algorithm: (**a**) the RMSEs of the DOA estimates; (**b**) the RMSEs of the CFO estimates.

**Figure 2 sensors-23-02115-f002:**
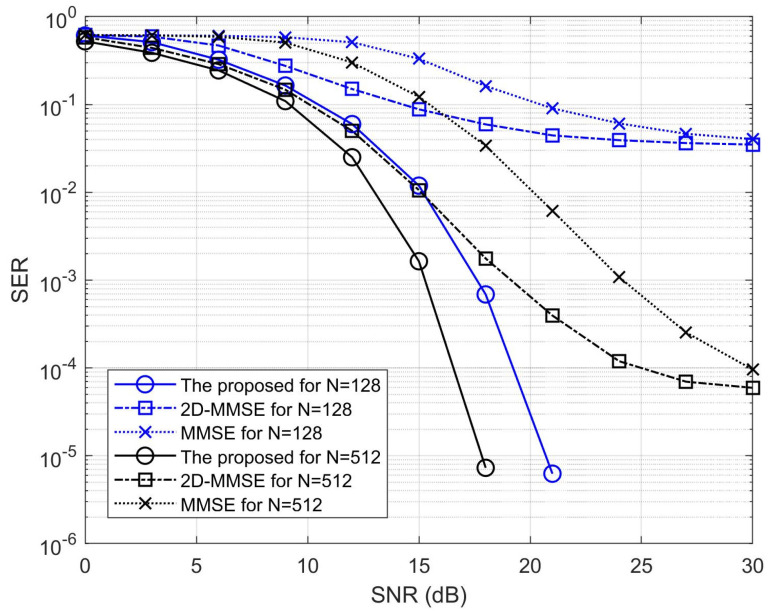
Comparisons of the SERs.

**Figure 3 sensors-23-02115-f003:**
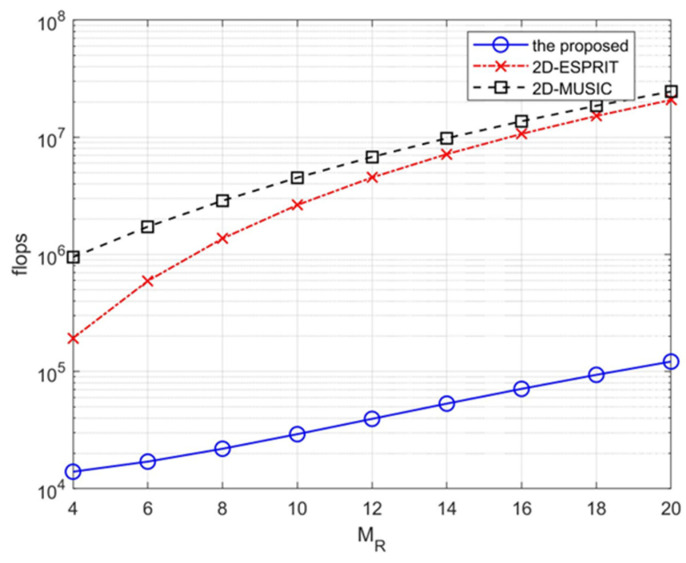
Comparisons of the computational complexities.

**Table 1 sensors-23-02115-t001:** The proposed F-ESPRIT algorithm for frequency bias estimation.

F-ESPRIT Algorithm	
Step 1:	Calculate the correlation matrix of ${\bar{y}}_{u,n}$: ${\bar{R}}_{u}=\frac{1}{N^{\prime }}\sum_{n=0}^{N^{\prime }-1}{\bar{y}}_{u,n}{\bar{y}}_{u,n}^{H},\;\forall u$
Step 2:	Find the largest eigenvector of ${\bar{R}}_{u}$, denoting as ${\tilde{v}}_{u}$.
Step 3:	Construct ${\tilde{v}}_{u,1}=J_{L,1}{\tilde{v}}_{u}$, and ${\tilde{v}}_{u,2}=J_{L,2}{\tilde{v}}_{u}$
Step 4:	Estimate the physical CFO of user by ${\hat{\omega }}_{u}=\frac{L}{2\pi }\arg\left\{{\tilde{v}}_{u,1}^{H}{\tilde{v}}_{u,2}\right\}.$

## Data Availability

Not applicable.

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
