# Peer review of "Efficient Space–Time Signal Processing Scheme of Frequency Synchronization and Positioning for Sensor Networks"

_sensors, 2023, doi:10.3390/s23042115_

Round 1

Reviewer 1 Report

This paper proposed an efficient space-time signal processing scheme of frequency synchronization and angle of arrival estimation for sensor networks. Unfortunately, the results showed in Fig. 1 presented that the proposed approach has degraded performance against conventional 2D-MUSIC scheme [10]. Since the effectiveness of the proposed method is obsolete compared to conventional scheme, the paper can not be accepted for publication due to lack of novelty. Also, the authors said that the proposed method is more efficient, numerical results to support that statement should be showed e.g. complexity analysis is desired. The quality of figures in the manuscript was also quite low.

Author Response

Comments:

This paper proposed an efficient space-time signal processing scheme of frequency synchronization and angle of arrival estimation for sensor networks. Unfortunately, the results showed in Fig. 1 presented that the proposed approach has degraded performance against conventional 2D-MUSIC scheme [10]. Since the effectiveness of the proposed method is obsolete compared to conventional scheme, the paper can not be accepted for publication due to lack of novelty. Also, the authors said that the proposed method is more efficient, numerical results to support that statement should be showed e.g. complexity analysis is desired. The quality of figures in the manuscript was also quite low.

Reply:

It is normal that 2D-based estimation algorithms generally exhibit a higher accuracy than that of 1D-based algorithms, because the former use a high dimensional signal vector contributing a higher processing gain compared to the later. However, the main disadvantage of conventional 2D-based algorithm is the considerably high computational complexity inherited from the usage of the high dimensional signal vectors.

The novelty of this study is to use two consecutive 1D-ESPRIT algorithms in conjunction with the beamforming process to estimate the DOAs and CFOs of the multiuser signals, substantially reducing the computational complexity. To illustrate this, in the revised manuscript we have discussed the detailed complexity of the proposed algorithm, and included a new figure (figure 3) to exhibit the superiority of the proposed algorithm in computational efficiency. Although the proposed algorithm has about 1.5 dB power loss in the DOA-CFO estimation as compared with the 2D-MUSIC algorithm, it has 1 dB power gain comparing to the 2D-ESPRIT algorithm, apart from a significantly lower computational complexity. Please see from line 159 to 167 of the revised manuscript for the analyses of computational complexity; and from line 196 to 206 for the discussions of new added figure 3.

In addition, all figures in the revised manuscript have been regenerated with an improved quality.

Reviewer 2 Report

In "Conclusions" Section, I would suggest citing numerical examples to show that 1D algorithm has higher estimation accuracy compared to 2D estimation technique.  Also, specify in terms of numerical values, how much better 1D algorithm compared to 2D.  It is important to specify, intuitively, why 1D is better compared to 2D.  Provide some possible extensions to the work presented. 

Author Response

Comments:

In "Conclusions" Section, I would suggest citing numerical examples to show that 1D algorithm has higher estimation accuracy compared to 2D estimation technique.  Also, specify in terms of numerical values, how much better 1D algorithm compared to 2D.  It is important to specify, intuitively, why 1D is better compared to 2D.  Provide some possible extensions to the work presented.

Reply:

  1. We have included a new figure (figure 3) in the revised manuscript, that compares the complexity of the proposed 1D-based algorithm to that of the 2D-based algorithm. Please see from line 159 to 167 of the revised manuscript for the analyses of computational complexity; and from line 196 to 206 for the discussions of new added figure 3.
  2. The conclusion has been revised as follows:

This study presents a low-complexity user positioning and frequency synchronization method for interleaved OFDMA uplinks. Compared with conventional 2D algorithms, the proposed 1D algorithm effectively mitigates the computational complexity and exhibits superior performance for MAI suppression, leading to at least 3 dB power gain in data detection, when SNR>12 dB. The proposed structured 1D-based algorithm is applicable to other high dimensional signal processing scenarios to provide a low-complexity solution, such as the 2D-DOA estimation of the signals received by a uniform rectangular antenna array.

Please see from line 209 to 215 of the revised manuscript.

Reviewer 3 Report

This paper presents a one-dimensional space-time signal processing scheme for the joint estimation of direction of arrival and carrier frequency offsets in OFDMA uplink systems. Overall the proposed method is proven effective and the manuscript is well-written. Some comments are as follows:

1. Different fonts are not used in the keywords.

2. The authors should add more research background in the Introduction, e.g., some brief introduction to the conventional 2D estimation algorithms and their pros and cons.

3. "According (20)" should be "According to (20)" in line 112 Page 5.

4. Figure 1 and Figure 2 can be improved with a higher resolution.

5. A new figure can be added with numerical evaluation for "3.4. Computational Complexity" in Page 6.

Author Response

Comments:

This paper presents a one-dimensional space-time signal processing scheme for the joint estimation of direction of arrival and carrier frequency offsets in OFDMA uplink systems. Overall the proposed method is proven effective and the manuscript is well-written. Some comments are as follows:

  1. Different fonts are not used in the keywords.
  2. The authors should add more research background in the Introduction, e.g., some brief introduction to the conventional 2D estimation algorithms and their pros and cons.
  3. "According (20)" should be "According to (20)" in line 112 Page 5.
  4. Figure 1 and Figure 2 can be improved with a higher resolution.
  5. A new figure can be added with numerical evaluation for "3.4. Computational Complexity" in Page 6.

Reply:

  1. We have correct the font inconsistency.
  2. We have included 2 new reference literatures, and have analyzed the pros and cons of conventional 2D-based algorithm in the last paragraph of the system model section of the revised manuscript. Please see from line 40 to 51 and from line 82 to 92 of the revised manuscript.
  3. Thank you.
  4. All the figures in the revised manuscript have been regenerated with an improved appearance.
  5. We have analyzed the detailed complexity of the proposed algorithm, and accordingly included a new figure (figure 3) in the revised manuscript, which compares the complexity of the proposed 1D-based algorithm to that of the 2D-based algorithm. Please see from line 159 to 167 of the revised manuscript for the analyses of computational complexity; and from line 196 to 206 for the discussions of new added figure 3.

Round 2

Reviewer 1 Report

The responses of the authors to the reviewer is fine and the revised manuscript is recommended for publication if the following minor issues can be resolved.

1) In the introduction part, the authors should add paragraph mentioning main contribution compared to conventional works, as replied to the reviewer. 

2) The format of the paper is quite poor. Some paragraphs were not aligned properly. In the reference list, there is one with author's name is underlined, that is different from the others. 

Author Response

Reply:

1)  We have included a new paragraph regarding the contribution of the proposed algorithm as follows:

"The aforementioned 2D-based estimation algorithms generally exhibit a higher accuracy than that of 1D-based algorithms, because the former use a high dimensional signal vector contributing high processing gain than the later. However, the main disadvantage of conventional 2D-based algorithm is the considerably high computational complexity inherited from the usage of the high dimensional signal vectors. The novelty of this study is to use two consecutive 1D-ESPRIT [11] algorithms in conjunction with the beamforming process to estimate the DOAs and CFOs of the multiuser signals, substantially reducing the computational complexity. To this end, we present a space-time signal processing scheme to jointly estimate the DOAs and CFOs of inter-leaved OFDMA uplink systems. The proposed approach involves using a 1D-ESPRIT algorithm to estimate the DOA of each user and then accordingly decomposing the received signal into a set of single user signals through a set of spatial beam-formers. The output of each spatial beamformer is used to estimate the CFO, which is automatically paired with the leading DOA of the spatial beamformer. According to the CFO estimates, the proposed approach compensates for the CFOs and then performs data detection by applying a reduced sized FFT on the output of the spatial beamformer, resulting in a significantly lower computational complexity than traditional 2D-based algorithms [8], [12]."

Please see from line 45 to 61 of the revised manuscript.

2)  We have checked the format of the revised manuscript according to the reviewer’s comment. 

Reviewer 3 Report

Dear authors,

Thanks for revising and resubmitting the manuscript. My previous concerns are solved.

Author Response

Thank you very much for your insightful comments.